# Graphene-like Carbon Structure Synthesis from Biomass Pyrolysis: A Critical Review on Feedstock–Process–Properties Relationship

Farhan Chowdhury Asif and Gobinda C. Saha *

Nanocomposites and Mechanics Laboratory (NCM Lab), University of New Brunswick, Fredericton, NB E3B 5A3, Canada
* Correspondence: gsaha@unb.ca; Tel.: +1-506-4587784

**Abstract:** Biomass pyrolysis is a promising route for synthesizing graphene-like carbon (GLC) structures, potentially offering a cost-effective and renewable alternative to graphene. This review paper responds to the call for highlighting the state of the art in GLC materials design and synthesis from renewable biomass microwave pyrolysis. This paper includes an introduction of the microwave pyrolysis technology, information on feedstock variability and selection, discussion on the correlation between microwave pyrolysis process conditions and pyrolyzed product characteristics, and, more importantly, a section identifying any differences between pyrolyzing feedstock using the microwave pyrolysis method vs. conventional pyrolysis method. Furthermore, this work concludes by detailing the knowledge currently missing with the recommendation for future research/innovation directions.

**Keywords:** graphene; graphene-like materials; carbon; biomass pyrolysis; microwave pyrolysis

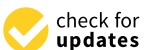



## 1. Introduction

In recent times, there has been a surge of enthusiasm for the utilization of renewable energy sources and the implementation of sustainable production approaches in order to overcome the forthcoming energy shortage and reach carbon neutrality [1–3]. As humanity's technological advances outpace the availability of resources, research endeavors are shifting towards the development of sustainable technologies that exploit novel materials with enhanced properties. Carbon, one of the most abundant elements on earth, has the potential to address this challenge. Traditional carbon materials, such as diamond, graphite, and carbon fiber, have been utilized for decades and have had a major role in economic and social progress [4]. The research in carbonaceous materials has evolved further since the discovery of carbon nanomaterials (CNMs) in the 1990s [5,6]. The distinctive and adaptable surface of carbon materials, along with their simplicity of production, makes them versatile targets and they are becoming increasingly important for use in the energy, biotechnology, biomedicine, and environment sectors.

Graphene, a single sheet of carbon atoms connected via $sp^2$ hybridization and arranged in a honeycomb lattice, has been widely regarded as one of the most revolutionary substances of the 21st century. Graphene has generated widespread interest in the scientific community since its discovery by Konstantin Novoselov and Andre Geim in 2004 [5], owing to its unparalleled characteristics [6]. Graphene is the world's first two-dimensional atomic crystal that has extraordinary characteristics, such as an electron mobility of $2.5 \times 10^5 cm^2 V^{-1} S^{-1}$ at room temperature, extremely high thermal conductivity above 5300 W/mK, a Young's modulus of up to 1 TPa, a tunable surface area of about 2675 $m^2$/g, an atomic thickness of ~0.335 nm, a density of 2200 Kg/$m^3$, ~97.7% optical transmittance, and an intrinsic strength of 130 GPa [7–12]. Theoretically, graphene has a higher electric double layer capacitance and specific capacitance than activated carbon, with 550 Fg$^{-1}$ and 268 Fg$^{-1}$, respectively, compared to activated carbon's 210 Fg$^{-1}$ [13].

In addition, this versatile material is also lightweight, impermeable to all gases, highly resilient to high current density, and easily amenable to chemical functionalization. The potential of graphene has yet to be fully explored, and its versatile characteristics present numerous opportunities for further research. Figure 1 shows a schematic representation of graphene properties.

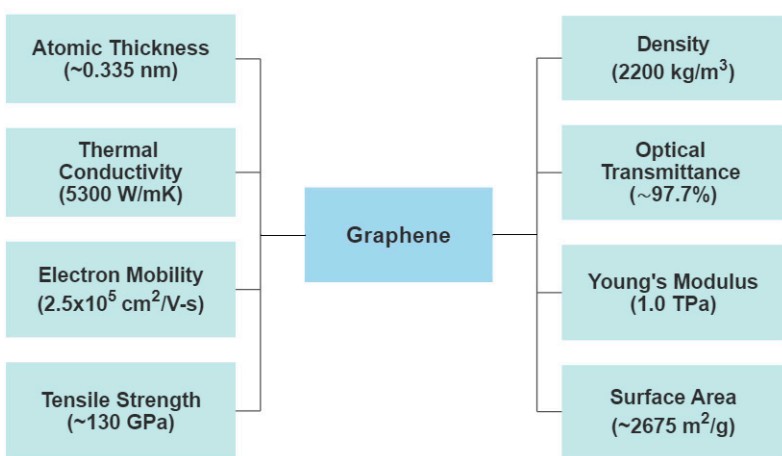

**Figure 1.** Schematic representation of graphene properties.

Graphene's honeycomb structure is the fundamental building block for the formation of other carbon allotropes, such as graphene oxide (GO), carbon nanotubes (CNTs), carbon nanodots, carbon nanoparticles, and fullerenes [14]. These allotropes differ structurally, with stacked honeycomb structures forming graphite, and rolled or wrapped honeycomb structures resulting in one-dimensional nanotubes and zero-dimensional fullerenes, respectively [15], as depicted in Figure 2a. Figure 2b–e represents the molecular structure of graphite, graphene, GO, and rGO.

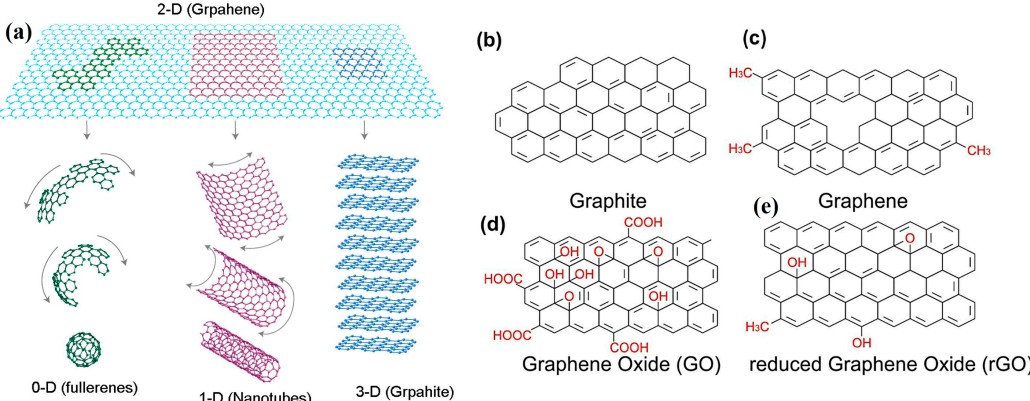

**Figure 2.** (**a**) Various carbon allotropes, such as 0D, 1D, and 3D, using 2-D graphene nanosheets as their basic building units (with permission from Ref [6]); the molecular structure of (**b**) graphite, (**c**) graphene, (**d**) GO, and (**e**) rGO [16].

The potential applications of graphene are far-reaching, covering areas such as photonics, composite materials and coating, energy generation and storage, environmental protection, biomedicine, and so on. Figure 3 depicts some of the important applications of graphene in research and industry.

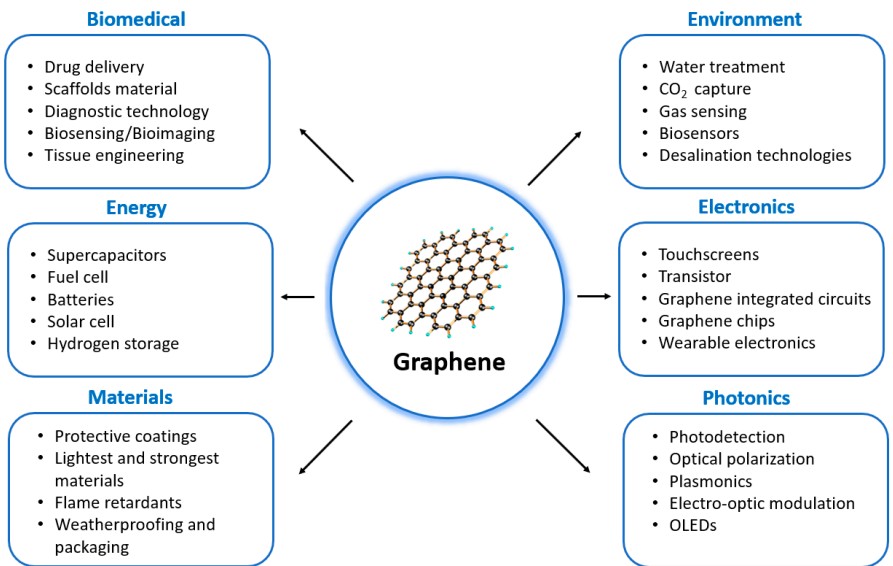

**Figure 3.** A diagram illustrating the various applications of graphene.

Graphene has been found to be an ideal material to improve the corrosion resistance of self-healing coatings due to its ability to enhance the mechanical properties, physical shielding ability, and energy conversion efficiency of the coatings [17]. For example, incorporating graphene into DA-epoxy resin composites has been shown to increase their self-repairing capacity and anti-corrosion performance [18]. Furthermore, graphene can also be used to tailor the flammability and flame retardancy of composite materials for safety applications. Taj et al. [19] investigated the flame properties of polymer-reinforced composites, with fillers consisting of nano-aluminum oxide and nano-graphene. It was determined that these fillers reduced flammability and improved flame retardancy, making them suitable for use in building materials. Tests were performed to analyze the burning rate, mass loss, and length loss of the composites, revealing that increasing the nano-graphene content up to 3% prevented flame propagation and resulted in a burning rate of zero.

Agrawal et al. [20] have developed a graphene-based filter to absorb $CO_2$ that met not only performance expectations for $CO_2$ collection but also recorded the highest $CO_2$ permeance. For instance, the filter exhibited a $CO_2$ permeance of 6180 GPU with a remarkable $CO_2/N_2$ separation factor of 22.5. This highlights the vital role that graphene can play in environmental applications. Additionally, a special form of graphene known as "Graph Air" is being employed in Australia to produce clean drinking water from highly contaminated water, with remarkable success compared to traditional filtration systems [21], demonstrating the versatility of graphene.

Graphene can have a significant positive impact on the energy sector too. Graphene Nanosheets derived from recycled plastics are being utilized in dye-sensitized solar cells (DSSCs) and supercapacitors, with DSSC yielding an impressive fill factor of 86.4% and a $V_{oc}$ of 0.77 V, and supercapacitors achieving a remarkable specific capacitance of 398 $Fg^{-1}$ [21,22]. In addition, the incorporation of highly conductive additives, such as graphene nanoparticles (GNPs), into phase change materials (PCMs) has been found to boost the thermal conductivity of these heat storage systems by as much as 220% when 3 wt.% GNPs are added to the PCM [23]. Graphene is also being investigated for potential use in smart wearable technology; for instance, researchers from Queen Mary University of London have developed a communicative piece of cloth utilizing graphene [24].

Graphene is also demonstrating its potential in the medical sector, with its antibacterial efficacy against both gram-positive and gram-negative bacteria being remarkable, and its high electrical and thermal conductivity making it a promising candidate in combating coronavirus [25]. In addition, smart drug nanocarriers (SDNCs) composed of chitosan

and nitrogen-doped graphene quantum dots (NGQDs) for pH-responsive drug delivery have been developed by Darwin et al. [26]. Using microplasma processing, it has been found that only 4.5% of the NGQD ratio is necessary for the SDNCs to become tough, thereby avoiding any exposure to high temperatures or hazardous chemical cross-linking agents. These composites can be used in biomedical applications and offer advantages such as strong drug-loading efficiency, pH-controlled sustained release of drugs, and stable solid-state PL properties for monitoring and therapeutic treatment. This study has opened new possibilities in the development of environmentally friendly and bio-compatible nanographene hydrogels with potential biomedical applications. Meanwhile, nano-graphene oxide (GO) is showing successful outcomes in orthopedic fields. According to the investigation conducted by Yitian et al. [27], 3D-printed biphasic calcium phosphate (BCP) scaffolds containing nano-GO have been identified to significantly enhance angiogenic effects as well as raise bone volume.

The food industry has also taken advantage of graphene-based materials for a variety of purposes, including aiding in plant growth, removing and detecting contaminants, and detecting patulin and quinolone [28]. Additionally, there are several articles discussing the application of graphene-based materials in medicine and biology [29], self-healing/protective coatings [30,31], smart drug/gene delivery [32], antimicrobial and coating applications in medicine and dentistry [33], strain sensors [34], catalysis [35], cryptography [36], electrosorption [37], desalination [38], electric vehicles, [39] as well as space technology [40].

The remarkable qualities of graphene have been well documented, but its utility is limited by the fact that its improved properties are only available with high purity. Moreover, the performance of graphene samples is affected by both the purity of the sample and the number of layers [15]. Thus, in the arena of graphene synthesis, the optimum outcome would be the mass production of graphene with purity levels equivalent to those produced by laboratory-scale synthesis.

The production of graphene and GLC materials can be performed using top-down and bottom-up methods. Top-down methods, such as mechanical exfoliation, chemical exfoliation, and chemical synthesis, are easy to apply for large-scale graphene production, but the quality of the graphene produced is often low. Bottom-up methods, such as chemical vapor deposition (CVD), epitaxial growth, and pyrolysis, are better for producing high-quality graphene with some structural defects and good electronic properties, though the amount produced is small. Defect-free, adjustable layer graphene can also be produced using bottom-up methods for special applications [41].

The CVD approach is highly cost-effective, but its yield is lower than other techniques and the removal of graphene from the metallic substrate is a complex operation. Moreover, this procedure can produce a considerable amount of hydrogen, which is a disadvantage. Mechanical exfoliation, on the other hand, presents a low yield output in overcoming the van der Waals force between the first and second layers without affecting the subsequent layers. This can have a detrimental effect on the performance of the devices due to the alteration of the 2D crystal's lattice structure. Additionally, the insolubility of macromolecules in organic syntheses, the emergence of unpredictable side effects with increasing molecular weight, and the inability of mechanical cleavage techniques to undergo mass production are all issues that must be addressed. Furthermore, liquid exfoliation techniques can yield low-conductivity graphene. Therefore, the development of a novel production process that can fulfill mass production and yield a superior end product is a subject of ongoing research [15].

Pyrolysis is a straightforward and well-liked thermochemical technique for synthesizing nanostructured carbon, which breaks down carbon sources into tiny pieces without oxygen. Historically, it has been employed in separate petroleum products but it is now utilized to convert waste agricultural materials into commercially and environmentally beneficial products. In recent years, microwave-assisted pyrolysis (MAP) of biomass has emerged as a promising pyrolysis method for minimizing time and energy, achieving

better heating efficiency, gaining greater control over the process, and producing more desired products than conventional pyrolysis [42]. However, the major challenge with this approach is the difficulty in regulating particle size, which leads to a wide range of diameters from 1 to 5 nm [43].

Biomass, composed of a high concentration of carbon and being a renewable resource, has been identified as a new potential source from which graphene and GLC materials can be derived. The amount of biomass waste produced around the world each year is estimated to be around 10 billion metric tons, and this number is anticipated to grow [44]. The advantages of biomass resources are their low cost and wide accessibility, which can potentially decrease the cost of graphene and other carbon-derived compounds. So far, several biomass materials had been effectively transformed into graphene and carbon-derived compounds, including wheat straw, sawdust, gumwood, bamboo, peanut shell, rice husks, sugarcane bagasse, orange peels, ginger, cotton, corncobs, and camphor leaves [45,46]. Biomass-derived graphene is frequently made up of aligned nanographene domains, which differ from ideal two-dimensional (2D) stacked graphene sheets and result in a variety of shapes, special functional groups, and amazing capabilities [47]. These carbons generated from biomass are frequently referred to as graphene-like materials.

In this review paper, we summarize and discuss some recent strategies to synthesize GLC material using solely the biomass pyrolysis process. We discuss the pyrolysis process and the correlation between different process parameters and pyrolyzed products, as well as the mechanisms of GLC material formation via biomass pyrolysis. Additionally, we provide a brief discussion on the special effect of microwave irradiation during pyrolysis on the morphology and microstructure of the pyrolyzed product.

## 2. Pyrolysis Process

Pyrolysis is a type of thermochemical conversion process that takes place without the presence of oxygen and is designed to break down the chemical bonds in a particular feedstock to decompose organic materials. This yields biochar, bio-oil, syngas, and other value-added products. The process is conducted in an oxygen-free environment with temperatures ranging between 400 °C and 1200 °C or even higher [48,49]. In this environment, biomass can be heated beyond its thermal stability limit without initiating combustion. Pyrolysis is a complex process involving various reactions and pathways such as depolymerization, dehydration, decarboxylation, intramolecular condensation, and aromatization, which take place at different temperatures and yield diverse product states for lignocellulosic components [49].

Depending on the heating mechanism used, pyrolysis can be classified into two categories: conventional pyrolysis (CP) and microwave-assisted pyrolysis (MAP). CP usually relies on an electric heating mechanism, which is often inefficient and energy intensive. MAP, on the other hand, has gained considerable attention from the research community due to its advantages over CP. MAP is faster, more energy-efficient, and offers greater precision over the process. Furthermore, MAP also results in higher heating rates and yields of desired products compared to CP [50–52]. Microwaves are a type of electromagnetic wave that falls between infrared and radio frequencies, with a frequency range of 300 MHz to 300 GHz and a wavelength that varies between 0.001 and 1 m. The majority of microwave reactors used in chemical synthesis, including those found in household kitchens, have a wavelength of 12.25 cm and a frequency of 2.45 GHz [53].

## 3. Microwave Pyrolysis Reaction Mechanism

In conventional pyrolysis, the heat is transmitted from an external source to the material's exterior and then to its core through conduction, convection, and radiation. Therefore, CP is inefficient, energy-consuming, and relies on convection and the thermal conductivity of the material being processed. In contrast, electromagnetic energy is transformed into heat energy in MAP. This occurs through microwaves entering the feedstock and then being stored as energy, which is then converted into heat inside the feedstock's core. This

method is advantageous because it avoids heat losses due to volumetric heating of the feedstock [53].

In MAP, the temperature of the biomass particle increases from the interior to the exterior, but in CP, it is the opposite. Furthermore, for both MAP and CP, the diffusion of volatile materials (mass flow) is always outward. Thus, heat flow and mass flow are concurrent for MAP and countercurrent for CP. Figure 4 shows the schematic of microwave and conventional heating methods. While the volatile elements diffuse from the interior core of the feedstock to its exterior surface, the surrounding of the feedstock is extremely hot during the CP process and relatively cooler for the MAP process. As a result of the improved heating mechanisms described above, MAP's heating and response mechanisms have significant advantages over CP. MAP offers many advantages, such as quick, precise, and even heating, saving time and energy, eliminating direct contact between the heat source and the material, transferring energy instead of heat, low thermal inertia, no need for prior treatment of the feedstock, quicker response time, better control, improved safety, and so on. Because MAP speeds up thermochemical processes and shortens reaction times, it also has the added benefit of reducing energy consumption [53,54].

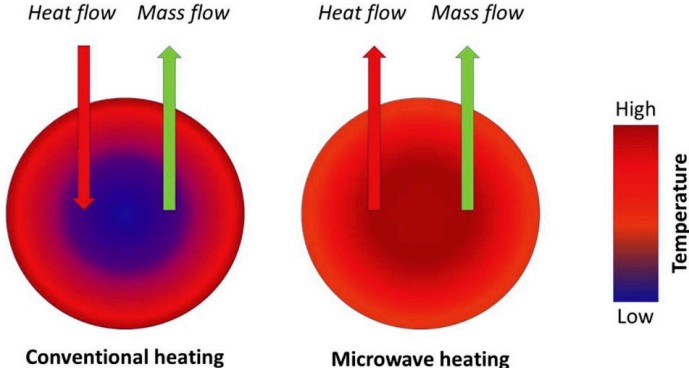

**Figure 4.** Illustration of the differences between microwave and conventional heating techniques. Reprinted from [54], with permission from Elsevier, Amsterdam, The Netherlands.

Also, it is worth noting that biomass does not absorb microwaves well, and therefore does not reach the necessary temperatures for pyrolysis when exposed to microwaves. Thus, pyrolysis systems employ external microwave absorption materials [55].

## 4. Key Distinction between MAP and CP

Robinson et al. [56] conducted a study that combined microwave pyrolysis, dielectric measurement, and fluid flow modeling to better understand the differences between microwave and conventional pyrolysis. Through their research, they were able to analyze and contrast the mechanisms of both processes. Their key finding was that the distinction is not between microwave and conventional heating, but rather between low and high heating rates. With low microwave power or domestic ovens, the heating rate can be comparable to conventional methods. When heating rates are low, vaporization of water within biomass structures is slow and pressure remains close to atmospheric. Pyrolysis in this case would proceed similarly to conventional methods with hemicellulose depolymerizing at temperatures over 200 °C, cellulose at 300 °C, and lignin in the 220–400 °C range with a diverse chemical composition produced. When heating rates are high (microwave heating), the vaporization rate increases, leading to pressure build-up, which elevates the boiling point of water remaining within biomass at temperatures well over 100 °C. This results in hydrolysis of hemicellulose at ∼130 °C, producing furfural as the primary product; hydrolysis of cellulose occurs at ∼175 °C, producing levoglucosan as the primary product; and lignin follows the same reaction scheme as conventional pyrolysis due to its lack of hydrolyzable linkages upon further heating, as illustrated in Figure 5.

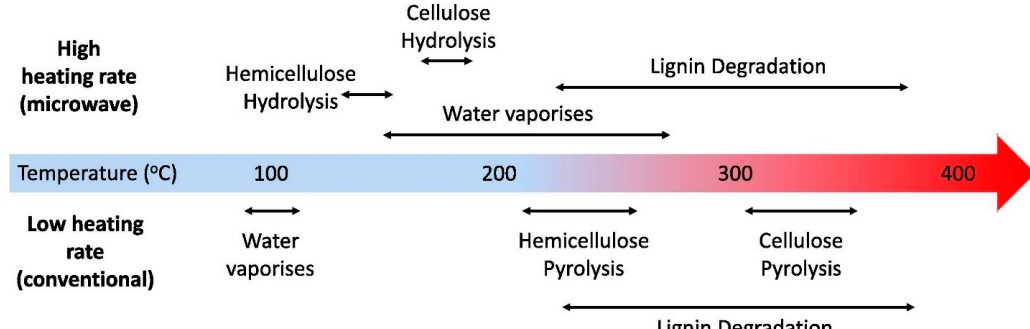

**Figure 5.** Scheme for MAP and CP. Reprinted from [56], with permission from Elsevier.

It was hypothesized that different behaviors with different biomasses occur not because of inherent chemistries but due to their micro- and macro-structures, which are characterized by permeability; high permeability would not sustain high pressures, so no difference between microwave vs. convectional pyrolysis would be expected. In contrast, low permeability does sustain those pressures, allowing for a difference in the mechanistic pathway based on the heating rate. The key finding of Robinson is also evident from the study of Zoraida et al. [57] who used anthracene oil as raw material for the production of carbon precursors via conventional and microwave technologies. In addition, they observed that the energy usage of microwave-assisted technology for producing these materials was up to 60% lower than traditional heating. The graphite obtained from microwave-based precursors had an excellent degree of graphitization, with smaller crystallite sizes than those from an exclusively microwaved coke. Graphene materials synthesized from these microwaved precursors showed improved lattice recovery, structure, and reduced oxygenated surface functional groups, especially when derived from microwave-derived coke.

## 5. Current Trends on Synthesis of GLC Materials via Biomass Microwave Pyrolysis Process

The process of creating GLC materials via microwave pyrolysis of biomasses has been explored by researchers in recent years. Even though they used various feedstocks and varied process settings, the basic process is nearly identical. This process involves three distinct steps: sample pre-treatment, pyrolysis, and post-treatment. During pre-treatment, the biomass sample is washed with deionized water to remove any contaminants, dried to eliminate moisture, and ground into a fine powder. Pyrolysis then takes place, followed by a post-treatment process that involves filtering, washing with deionized water, and drying. In this section, we have highlighted some of the recent studies on GLC materials synthesis through the biomass pyrolysis process.

Zhang et al. [58] investigated the synthesis of hollow carbon nanofibers (HCNFs) on the surface of biochar, using pine nutshell (PNS) as feedstock, without the use of a catalyst. The PNS was crushed until it had a particle size of 65–200 μm and then put into a vacuum oven and heated to 105 °C for 10 h to take out the moisture. Afterward, it was blended with commercially accessible biomass-based activated carbon (AC) in an 8:2 mass ratio. The samples were then pyrolyzed for 20 min at 400 °C, 500 °C, 600 °C, and 700 °C in a $N_2$ atmosphere. The results indicated that 600 °C was the optimal temperature for the synthesis of the HCNFs, resulting in the formation of fewer organic matrixes, functional groups, structural defects, and imperfections. It was proposed that pyrolysis volatiles forced their way out of surface pores, solidified, and graphitized the vapor on the biochar exterior, leading to the formation and development of HCNFs. On the other hand, Gopalakrishnan et al. [59] developed a simple one-step method to synthesize few-layer graphene-like porous carbon nanosheets (FLG-CNs) using ginger as a feedstock. The initial step of the process was to slice up the fresh ginger and rinse it with deionized water to eliminate any external impurities. This was followed by oven drying, and then pyrolyzing the ginger at temperatures of 600 °C, 800 °C, and 900 °C for an hour under an argon gas

atmosphere. Since ginger contains a variety of minerals, when it is heated, these minerals become porous and increase the surface area, which is ideal for the performance of electric double-layer capacitors (EDLCs). It was validated by the electrode constructed of FLG-CNs at 800 °C exhibiting an excellent specific capacitance of 390 Fg$^{-1}$ when the current density was 1 Ag$^{-1}$, and the capacity remained at 93.3% even after 3500 charge/discharge cycles.

Researchers have utilized different biomasses to synthesize graphene oxide (GO). Danafar et al. [60] demonstrated the synthesis of nano-sized GO flakes from onion sheaths through pyrolysis coupled with sonochemistry. To remove the surface dust, onion sheathings were washed with deionized water and air dried. Then, they were pyrolyzed at 700 °C under N$_2$ gas flow. The obtained GO-like carbon flakes were washed with a combination of water and ethanol in order to eliminate any remaining byproducts, then sonicated in deionized water to convert them into nano-sized flakes (6.6 ± 2.4 nm) of a uniform size. Figure 6 illustrates the X-ray diffraction (XRD) patterns of both nano-sized GO-like carbon flakes and pyrolyzed GO-like flakes, where an intensification in the intensity of the carbon flakes following ultrasound treatment was noticed, implying the reinforcement of the stacking arrangement of the aromatic layers in the nanoflakes.

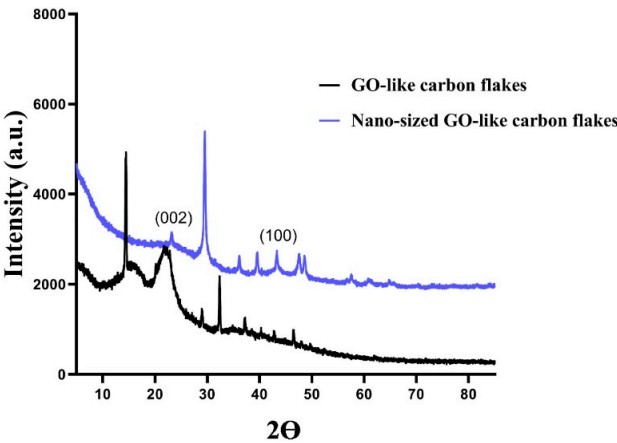

**Figure 6.** The XRD patterns of both pyrolyzed and nano-sized GO-like carbon flakes. Reprinted from [60], with permission from Elsevier.

A straightforward and economic procedure was developed by Somanathan et al. [61] to synthesize GO from agricultural waste. For this study, sugarcane bagasse was first crushed and grounded to produce a powder. A muffle furnace was used to heat 0.5 g of powder and 0.1 g of ferrocene for 10 min at 300 °C under atmospheric conditions. They found that the produced GO presented a well-graphitized structure. Another study was carried out by Hashmi et al. [62] using orange peel (OP), sugarcane bagasse (SB), and rice bran (RB) both individually and as tri-composite agro-waste (TAW) mixtures to produce GO. They prepared the feedstock by washing it with water and then drying it in the sun. The dry feedstocks were then processed using a mortar, pestle, and mixer to create fine powder. For preparing GO from the individual agro-waste, they mixed 0.3 g of each feedstock powder with 0.1 g of ferrocene. The mixture was then heated in a muffle kiln at 300 °C for 15 min. To prepare GO from the tri-composite mixture, they mixed half a gram of each feedstock powder with 0.3 g of ferrocene and then heated the mixture in a muffle kiln at 400 °C for 15 min.

Figure 7 is the illustration of the XRD spectrum of their study. From the XRD patterns, they observed that GO was successfully prepared only by TAW with the main diffraction peak located at 2θ = 12.705. They also observed excellent crystallinity that was indicated by the prominent and strong peak of GO (Figure 7). Debbarma et al. [63] also synthesized graphene oxide from sugarcane bagasse using pyrolysis at a low temperature ranging from 250 °C to 450 °C. The first step in the process was to chop the sugarcane bagasse into tiny pieces then wash it with deionized water to get rid of any contaminants. The

samples were left to dry in the sun for a few days before being heated in an oven at 70 °C for a period of 24 h. Afterwards, the samples were ground into a fine powder and subjected to pyrolysis at 250 °C, 350 °C, and 450 °C for durations of 1 h, 30 min, and 10 min, respectively. After this, the material that resulted from pyrolysis was filtered, rinsed with warm, deionized water, and left to dry for 24 h. In this study, it was observed that 350 °C was suitable for condensation and aromatization of the glucose monomers to form graphene oxide nanosheets at a large scale. In another study, this same research group synthesized graphene nanosheets following the same process mentioned above in the presence of sodium hydroxide [64] and they found that mixing sodium hydroxide with it in a 1:1 ratio prevented oxygen from attacking the sample during the pyrolysis process.

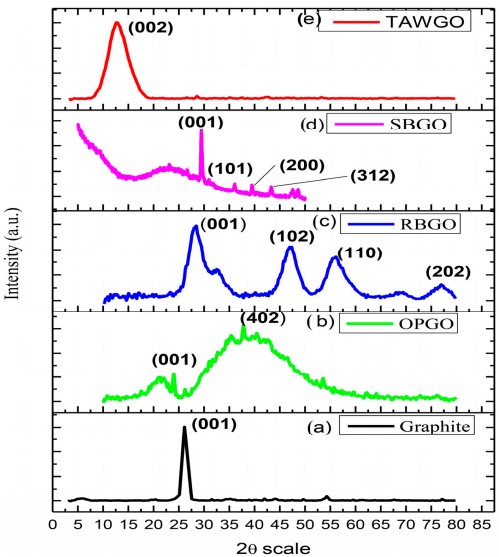

**Figure 7.** XRD spectrum of (**a**) graphite; and GO synthesized from (**b**) OP, (**c**) RB, (**d**) SB, and (**e**) TAW [62], reprinted with permission from the publisher (Taylor & Francis Ltd., Abingdon, Oxfordshire, England, http://www.tandfonline.com (accessed on 20 January 2023)).

Liu et al. [65] and Wang et al. [66] synthesized biobased graphene by pyrolyzing kraft lignin (KL) and bamboo biomass, respectively. Liu prepared biobased graphene in the presence of iron catalyst using commercial KL as feedstock. In this experiment, five grams of KL was mixed with Fe powder of varying ratios (1:2, 1:3, 1:5, and 1:7) before being heated in a quartz tube at 1000 °C under an argon flow. The samples were kept in the quartz tube for periods of 60, 75, 90, 105, and 120 min. Once the thermal treatment was finished, the sample was cooled back to room temperature and rinsed multiple times with deionized water. The iron particles were isolated by means of magnetic separation and any remaining iron was eliminated by washing the specimens with 10% hydrochloric acid. It was observed that the thermal treatment process lasting 90 min, with a ratio of 3:1 of carbon source to iron, resulted in graphene of superior quality. Moreover, carbon nanotubes (CNTs) were seen when the thermal treatment lasted 105 min. In contrast, Wang et al. [66] synthesized graphene-containing biochar from waste bamboo biomass with the activating agent $K_2CO_3$ and the help of microwave-assisted catalytic graphitization. The process of this study began by washing the raw bamboo with deionized water to clear away any contaminants, followed by drying and reducing it to particles that were smaller than 0.1 mm. It was then carbonized at 400 °C while under a flow of nitrogen gas for a period of three hours. The carbonized material was combined with potassium carbonate at a ratio of 1:3, and pyrolyzed at 900 °C under a nitrogen atmosphere for 25 min. For the post-treatment, the sample was given multiple rinses with deionized water and had its pH adjusted by adding a weak solution of hydrochloric acid. Lastly, the sample was dried at 105 °C for 12 h. The resulting biochar exhibited a typical graphene structure, plenty of micropores, and a huge

surface area of up to 1565 $m^2g^{-1}$. The activating agent $K_2CO_3$ played a significant role in facilitating transformation of amorphous carbon into graphene-like carbon.

CNTs were also synthesized by different researchers using different biomasses and varied pyrolysis processes. Yu et al. [67] developed a new approach that uses microwave pyrolysis for producing super-long carbon nanotubes (SL−CNTs) without the need for an outside catalyst. By pyrolyzing cellulose at temperatures ranging from 1200 to 1400 °C, they were able to generate CNTs with lengths between 0.7–2 mm. For this study, cellulose was derived from a palm kernel shell (PKS) and AC was adopted as the microwave absorbing material. The cellulose sample was first oven dried for 12 h at 80 °C and then mixed with AC at the ratio of 10:2. Next, the specimen was placed inside a quartz tube and pyrolyzed at 600 °C for a period of 30 min under $N_2$ environment. In the second step, 5 g of the produced char from the first step pyrolysis was again pyrolyzed at 1200 °C, 1300 °C, and 1400 °C for 30 min under $N_2$ environment. As the pyrolyzing temperature increased, they observed that the average length of CNTs increased. They also observed that CNTs' shapes changed from twisted, coiled, and threadlike to straight structures as the temperature varied. Furthermore, they noticed that the carbon order in the SL−CNTs increased after microwave treatment at 1400 °C, which was evidenced by the low Raman $I_D/I_G$ ratio of 0.84. Additionally, the inorganic components found in the biomass were thought to act as a catalyst, accelerating the growth of the SL−CNTs. These observations are depicted in Figure 8.

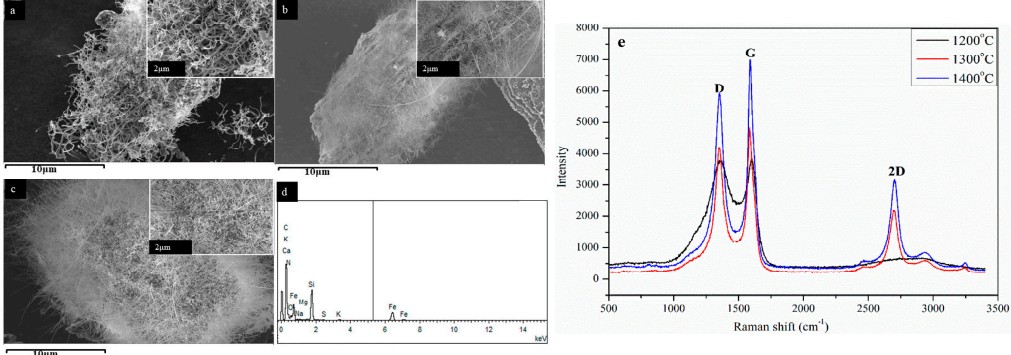

**Figure 8.** SEM images of SL-CNTs following microwave treatment at temperatures of: (**a**) 1200 °C, (**b**) 1300 °C, and (**c**) 1400 °C; (**d**) EDS spectra of SL−CNTs synthesized at 1300 °C; (**e**) Raman spectra of SL−CNTs at different temperatures. Reprinted from [67], Copyright 2022, Esohe Omoriyekomwan J, Tahmasebi A, Zhang J, Yu J., used under Creative Commons Attribution License (CC BY) (https: //creativecommons.org/licenses/by/4.0/ (accessed on 20 January 2023)).

Hidalgo et al. [68] developed another method for synthesizing CNTs from biochar using microwave irradiation and ferrocene as a catalyst. The first step in this process was to pyrolyze 100 g of agro-industrial residual biomass containing wheat straw, rapeseed cake, oat hulls, and hazelnut hulls at temperatures of 400 °C and 600 °C under $N_2$ gas flow for 3 hrs. The produced biochar was then separated, milled, and fractioned using a sieve (size < 75 μm). In the second step, the biochar was mixed with ferrocene and pyrolyzed at 80 °C for 5 min. Researchers observed that CNTs were present in higher concentrations when biochar was pyrolyzed at 600 °C. Furthermore, a superior quality of CNTs with a higher rate of graphitization was observed when biochar created from hazelnut hull and wheat straw was employed.

Researchers have also synthesized graphene sheets using a biomass pyrolysis process. Xia et al. [69] synthesized three-dimensional porous graphene-like sheets (3DPGLS) with an impressive level of purity, negligible defect rate, great electrical conductivity, and a large specific surface area (1506.19 $m^2g^{-1}$). For this experiment, coconut shells were ground to a particle size of less than 100 μm and then carbonized at a temperature of 400 °C for three hours in a nitrogen atmosphere. The biocarbon powder created from the carbonization

process was blended with $K_2CO_3$ in a ratio of 1:2 and pyrolyzed at 900 °C in a nitrogen atmosphere for 2 h. Lastly, the samples were cooled down, followed by a treatment of dilute hydrochloric acid and de-ionized water washes, and dried for 12 h at 60 °C. Widiatmoko et al. [70] also synthesized graphene sheets through a two-step pyrolysis process using oil palm empty fruit bunch (EFB). In the first step, they ground the EFB into powder using a ball mill and blended it with 3M $FeCl_2$ as a catalyst, with urea as a nitrogen source (1:1) and with $ZnCl_2$ as an activator (2:1). Then the sample was heated on a hot plate at 80 °C for 2 h, followed by 2 h in the oven for drying. For the first stage pyrolysis, the samples were pyrolyzed at 250 °C, 350 °C, and 450 °C for a duration of 60 min. The second stage involved raising the temperature to 900 °C and maintaining it for an additional 90 min. It was observed that the yield of the pyrolysis product was influenced by the temperature used in the initial pyrolysis step. Furthermore, it is a well-known fact that lignin yields more aromatics and char than cellulose when pyrolyzed, and that decomposition of lignin is most effective in the 350–450 °C range. Widiatmoko's study showed that, when pyrolysis began at a temperature that corresponded to the decomposition temperature of lignin, it resulted in a great deal of graphene with a very high carbon concentration. There are also other studies on synthesizing GLC materials via pyrolysis. A brief summary is given in Table 1.

**Table 1.** GLC materials synthesis via pyrolysis from different bio sources.

| Biomass Sources | Reaction Temperature | Pyrolysis Environment | Retention Time | Catalyst | Pyrolyzed Product | Ref. |
|---|---|---|---|---|---|---|
| Waste Tea | 800 °C | $N_2$ gas | 1 h | Potassium Ferrate | Multi-hierarchical porous carbon | [71] |
| Peanut Shell | 800 °C | $N_2$ gas | 1 h | | | |
| Pomelo Peel | 800 °C | $N_2$ gas | 1 h | | | |
| Spent Tea | 1st stage: 1000 °C 2nd stage: 100–900 W | Inert | 3 h 15–180 min | $HNO_3$ | Graphene quantum dots | [72] |
| Quercus ilex leaves | 820 °C | - | 3 h | ZSM-5 and bentonite clay | Metal-doped graphene sheets (MDGs) | [73] |
| Waste biomass-derived cellulose | 800 °C | $N_2$ gas | 2 h | KOH | Multilayered graphene | [74] |
| walnut shell | 850 °C | Ar gas | 90 min | KOH | Graphene-like (GL) porous carbon | [75] |
| Dried green tea leaves | 900 °C, 1100 °C | $N_2$ gas | 3 h | - | Few-Layer Multifunctional Graphene | [76] |
| Chitosan | 600 °C–800 °C | Ar gas | - | - | N-doped graphene | [77] |
| Biomass guanine | 1000 °C | $N_2$ gas | 4 h | - | GL 2D carbon | [78] |
| Gumwood | 500 °C | $N_2$ gas | 30 min | - | CNTs | [79] |
| Okara | 800 °C | $N_2$ gas | 2 h | - | N-doped GL mesoporous nanosheets | [80] |

## 6. Effect of Microwave on GLC Materials Synthesis via Pyrolysis

In the process of GLC materials synthesis via pyrolysis, the choice of heating method plays a key role in determining the amorphous phase transition of graphene. Direct pyrolysis via traditional heating methods typically results in an amorphous phase and small graphite clusters. On the other hand, microwave heating can be used to convert electromagnetic energy into heat energy at the molecular level, leading to the creation of localized hotspots with a much higher temperature than the bulk material. These hotspots act as nucleation sites, which promote the rearrangement of molecules from an

unordered phase to a crystalline phase. Furthermore, microwave radiation treatment can be used to transform $sp^3$ bonds to $sp^2$ bonds, in preparation for graphene formation [81]. Additionally, this method can improve the degree of graphitization of carbon material at a lower temperature in a shorter period without the need for a catalyst, while also producing a higher yield of few-layer graphene compared to conventional heating techniques [82]. A better understanding of the effect of microwave on the synthesis of carbon-based materials could be gained from the studies of Omoriyekomwan et al. [83] and Kaiqi et al. [79].

Omoriyekomwan et al. [83] conducted a study where they compared the results of forming hollow carbon nanofibers (HCNFs) via microwave pyrolysis of palm kernel shells at 500 °C and 600 °C to those synthesized with fixed-bed pyrolysis. They observed that the development of HCNFs could only be detected during microwave pyrolysis, implying that microwave radiation played an important role in the production of these nanostructures. This growth was believed to be due to the microwave radiation being absorbed by the biomass, causing an electric arc formation and devolatilization. The heavy components of the volatile matter then resolidified on the surface as a result of lower temperatures, forming carbon nanospheres. These nanospheres then self-extrude outward from the biomass particle through nano-sized channels, initiating HCNF growth, which is known as the "self-extrusion model growth". In comparison, when conventional heating was used in fixed-bed pyrolysis, the surrounding temperature was higher than the particle core, preventing volatiles from solidifying and undergoing secondary cracking instead.

Kaiqi et al. [79] synthesized multi-walled CNTs via microwave-induced pyrolysis of gumwood. In this experiment, gumwood was pyrolyzed at 500 °C and maintained for 30 min under an oxygen-free atmosphere with nitrogen gas flowing at a rate of 100 mL/min. The gumwood was then combined with SiC in a 20:1 mass ratio, and the resultant compounds were separated for further analysis. For comparison, conventional pyrolysis was also performed at the same temperature and nitrogen flow rate. In terms of morphology and microstructure, the researchers observed that the chars formed by microwave-induced pyrolysis differed from those produced by conventional pyrolysis, with the latter having no CNTs on their surfaces. They attributed the formation of CNTs under microwave-induced pyrolysis to the special effect of microwave radiation on the thermochemical processing of biomass. They proposed a mechanism for CNT development under microwave irradiation, in which volatiles released from biomass formed char particles, which then served as substrates. Mineral matter in char particles served as a catalyst, while released volatiles served as a carbon source gas, undergoing thermal and/or catalytic breaking on char particle surfaces. As a result of the impacts of microwave irradiation, amorphous carbon nanospheres formed, which then self-assembled into multi-walled CNTs. This method had the benefit of producing localized hot spots that could graphitize CNTs at far lower temperatures than conventional heating. The advantage of this approach was that it efficiently produced localized hot spots that graphitized CNTs at much lower temperatures than conventional heating requires.

## 7. Suitable Biomass Feedstock for GLC Materials Synthesis via Pyrolysis

Recently, there has been an increased focus on the synthesis of graphene or GLC materials from various biomass sources due to their sustainability, non-toxicity, environmental friendliness, cost-effectiveness, and ease of acquisition. Lignocellulosic biomass, composed primarily of lignin, cellulose, and hemicellulose, is particularly attractive due to its potential to produce higher-quality GLC materials with greater surface areas. Though the influence of feedstock composition on GLC material synthesis is not well understood, Table 2 suggests that feedstocks with higher carbon contents are more suitable for producing GLC materials via pyrolysis. Additionally, the characteristics of the final product largely depend on the synthesis method rather than the feedstock material. Table 2 provides elemental and proximate analysis of various bio precursors that have been used to synthesize GLC materials via various methods.

**Table 2.** Elemental and Proximate Analysis of various bio precursors.

| Biomass Sample | Proximate Analysis, wt.% | | | | Ultimate Analysis, wt.% | | | | Ref. |
| --- | --- | --- | --- | --- | --- | --- | --- | --- | --- |
| | Moisture Content | Volatile Matter | Fixed Carbon | Ash | C | $H_2$ | $N_2$ | $O_2$ | |
| Softwood | 11.5 | 67.3 | 19.5 | 1.7 | 44.43 | 6.16 | 0.18 | 49.23 | [84] |
| Hemp | 10.7 | 69.6 | 18.8 | 0.9 | 45.71 | 5.89 | - | 48.40 | |
| Rice straw | 8.25 | 72.20 | 14.44 | 13.36 | 45.41 | 6.28 | 0.99 | 47.11 | [85] |
| Pine nutshell | 2.12 | 74.53 | 22.63 | 0.94 | 50.16 | 5.81 | 0.28 | 43.41 | [58] |
| Palm Kernel Shell | 14.90 | 74.68 | 23.68 | 1.64 | 49.90 | 5.25 | 0.36 | 43.54 | [83] |
| Populus wood | - | - | - | - | 39.75 | 6.09 | 1.52 | 52.54 | [86] |
| Spent Coffee Beans | - | - | - | - | 49.30 | 3.61 | 2.24 | 41.33 | [87] |
| Rice husk | 6.81 | 59.8 | 13.68 | 19.71 | 40.71 | 4.97 | 0.49 | - | [88] |
| Sugarcane bagasse | 9.51 | 74.98 | 13.57 | 1.94 | 43.77 | 6.83 | - | 47.46 | [89] |
| Orange Peel | - | - | - | 3.05 | 49.59 | 6.95 | 0.66 | 39.7 | [90] |
| Chitosan | - | - | - | - | 45.65 | 7.66 | 7.6 | 39.09 | [91] |

The lignocellulosic contents of several bio precursors previously employed to synthesize GLC materials are shown in Table 3. Each component of lignocellulosic biomass decomposes differently, and the breakdown is affected by temperature, heating rate, and the presence of contaminants [92]. The three components disintegrate at various temperatures, with hemicellulose being the one that would pyrolyze the most easily. Due to the intricate structure and higher resistance to high temperatures than hemicellulose and cellulose, lignin would be the most challenging to pyrolyze [93]. Studies on the influence of cellulose, hemicellulose, and lignin content on the formation of GLC materials via pyrolysis are scarce. According to earlier research, at higher temperatures, cellulose develops significant assemblages of polycyclic aromatic hydrocarbon domains, but lignin and hemicellulose only produce a small amount. That is why early experiments investigated the generation of graphitic carbon from pure cellulose [94]. Additionally, according to a few studies [95,96], lignin is the component of lignocellulosic biomass that is most conducive to the synthesis of laser-induced graphene (LIG). Therefore, future studies may concentrate on the impact of various lignocellulosic components on the quality and characteristics of the GLC materials produced through the pyrolysis process.

**Table 3.** Lignocellulosic Content of various bio precursors.

| Biomass | Cellulose (wt.%) | Hemicellulose (wt.%) | Lignin (wt.%) | Ref. |
| --- | --- | --- | --- | --- |
| Hemp | 53–91 | 4–18 | 1–17 | [97] |
| Rice Husk | 32.67 | 31.68 | 18.81 | [98] |
| Sugarcane Bagasse | 50 | 25 | 25 | [99] |
| Empty Fruit Bunches of Palm Oil | 37.26 | 14.62 | 31.68 | [100] |
| Wheat straw | 34.40 | 20–25 | 20 | [101] |
| Palm Kernel Shell | 27.7 | 21.6 | 44 | [102] |
| Bamboo | 47.2 | 23.9 | 25.3 | [103] |
| Rice Straw | 29.2–34.7 | 12.0–29.3 | 17.0–19.0 | [104] |
| Switch Grass | 30–50 | 10–40 | 5–20 | [105] |
| Miscanthus | 24 | 44 | 17 | [106] |
| Walnut Shell | 23.9 | 22.4 | 50.3 | [107] |

### 8. Correlation between Microwave Pyrolysis Process Conditions and Pyrolyzed Product Characteristics

Microwave pyrolysis depends on the interaction between feedstock and microwave irradiation. The quality and features of the product obtained from microwave pyrolysis of biomass mostly rely on the operational conditions and the properties of biomass feedstocks. In this section, we explore the relationship between the various process parameters of pyrolysis and their effect on the final pyrolyzed product, with the goal of providing insight into how to optimize the production of GLC materials.

The performance of microwave pyrolysis is greatly affected by the microwave power, particle size, and batch size of the feedstock. Figure 9a illustrates that increasing the microwave power increases both the heating rate and the maximum reaction temperature. Additionally, reducing the particle size also increases both the heating rate and maximum reaction temperature, as shown in Figure 9b. This can be attributed to increased bulk density and intra-particle contact area with reduced particle size [85]. Furthermore, reducing the particle size to a specific size (e.g., less than 0.25 mm) has been observed to prevent heat from transferring within the particles, thus slowing the process of pyrolysis [108]. Parthasarathy et al. [109] found that increasing the feedstock particle size increases both the char yield and carbon content of the char. Lastly, the batch size of the raw material has been found to play a crucial role in microwave pyrolysis. Figure 9c shows that using a small amount of starting material (5–15 g) yields higher heating rates and higher residence temperatures at a much lower microwave power [110,111].

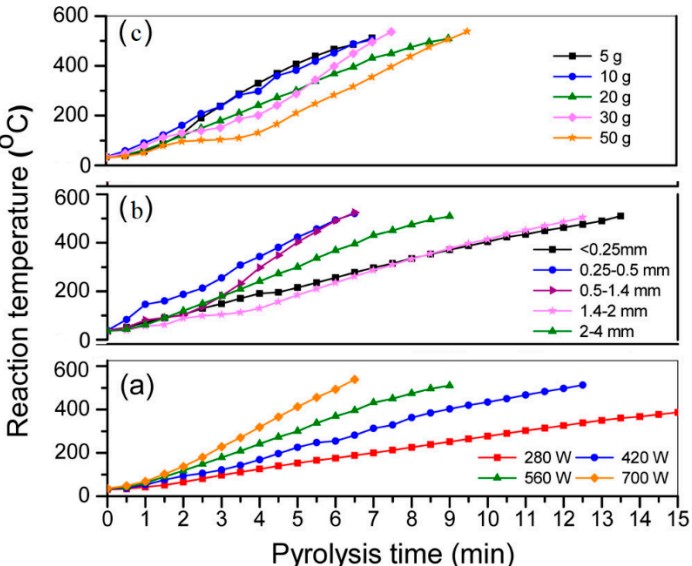

**Figure 9.** Reaction temperature profiles for various circumstances of microwave pyrolysis: (**a**) microwave power effect, (**b**) effect of particle size, (**c**) effect of initial mass. Reprinted with permission from [108]. Copyright 2015 American Chemical Society.

Low initial moisture levels in biomass are usually preferable for the pyrolysis process. Drying the biomass before subjecting it to pyrolysis improves the energy efficiency of the process [112]. Demirbas observed that increasing moisture content decreases biochar yield and increases the yield of liquid product [113]. However, a few studies have found that increasing the moisture level of the feedstock leads to an increase in char and gas generation [114–116]. Furthermore, the reaction temperature decreases as the moisture content increases, because when the mixture is heated up, heat is expended in the process of drying out the components [117]. Darmstadt et al. [118] observed that feedstock moisture content had a greater impact on softwoods than hardwoods. In addition, Xiaodi Li et al. [119] investigated the effects of preheat temperature on pyrolysis properties and product properties, finding that preheating the feedstock before microwave pyrolysis increased biochar

yield, shortened the initial time for rapid temperature rise, and increased the BET surface area. The effect of thermal pretreatment was also reported by Jian et al. [120]. Feedstock moisture content also influences the heating rate. A lower heating rate arises from higher moisture content [121]. Additionally, it was noted that as initial moisture content increased, the specific char surface area increased, though the effect was more pronounced at lower pyrolysis temperatures [122]. Furthermore, it is common practice to dry biomass feedstock prior to pyrolysis, and the feedstock is frequently dried to a moisture level of less than 10% [123].

The reaction temperature of pyrolysis has a significant impact on the char yield and characteristics. According to Mohammad et al. [124], the char yield reduces as the pyrolysis temperature rises. At higher temperatures, the devolatilization process accelerates, which leads to more vapors and gases being produced and a decrease in the char yield [125]. With the help of Raman spectroscopy, Asadullah et al. [126] observed that an increase in temperature causes char to aromatize more quickly. The maximum pyrolysis temperature also affects the surface area, pore structure, and carbon content. According to one study, the BET surface area of a char decreases as the pyrolysis temperature rises. This was found to be quite drastic, with the surface area dropping by a minimum of 200 times when the temperature was raised from 500 to 800 °C. It is believed that the drastic reduction in micropores which occurs between 500 °C and 800 °C is what caused the sharp decline in the surface area [122]. In a different analysis, Fu et al. [127] observed that the surface area of char increases with temperature; however, it decreases slightly if the temperatures surpass 1173 K. Zhao et al. [128] also found that pyrolyzing the rapeseed steam from 200 to 700 °C resulted in an increase in surface area, from 1 to 45 $m^2/g$, which shows more of a carbonaceous, aromatic structure for the biochar. This behavior can be attributed to the release of volatile gases and the formation of pores at higher temperatures. Furthermore, Lua et al.'s [129] research indicates a general correlation between the BET surface area, micropore surface area, and total pore volume, which increases up to a specific temperature and then begins to decrease gradually. The initial increasing trend could be attributed to the emission of low-molecular-weight gases from the carbon structure. The declining trend might be related to the weakening and liquifying of some of the residual volatiles in the char, which causes an intermediate melt to form in the chars. This intermediate melt obstructs the formation of the char's primitive pore structure by partly sealing some of the pores. Nevertheless, when the pyrolysis temperature was increased further, it caused the pores to grow and develop, thus leading to an increase in the BET surface area, the micropore area, and the total pore volume. The char's intermediate melt undergoing depolymerization and evaporating is the reason behind this phenomenon, which causes the previously sealed pores to open up, as well as the formation of new pores as a result of the disappearance of the heavier volatiles. As the pyrolysis temperature increased further, decreases in BET surface area, micropore surface area, and total pore volume were observed. This was likely due to the compression of pores within the char and the narrowing of the pore openings, which both cause the accessible pore surface area to be reduced. Another potential factor might be the production of secondary melt from high-molecular-weight volatiles, comparable to the previously stated intermediate melt. The fixed carbon content and carbon content of char increase with temperature as a result of deoxygenation and dehydration, indicating greater structural ordering for lowering reaction site concentration [128,130,131]. Furthermore, with increasing pyrolysis temperature, the total volume of pores increases, but the average pore diameter decreases as a result of an increase in the proportion of relatively tiny pores [128].

The char characteristics and yield are significantly Influenced by the heating rate as well. According to Mohammad et al. [124], the char yield reduces as the heating rate rises. Although, the influence of the rate of heating was more noticeable at higher temperatures when it came to the production of char [132]. However, compared to char produced at low heating rates, high-heating-rate char has a smaller surface area [127]. It is believed that this is due to an excessive heating rate that raises the temperature of the char interior and results

in partial graphitization and the construction of a graphene structure: neither of which contributes to the development of a large surface area. On the contrary, Zhao et al. [128] found that rapeseed stem surface area increased at first with the increasing heating rate due to a larger extent of thermal decomposition and then slightly decreased. When the heating rate is increased, the carbon content of char decreases slightly while the hydrogen and oxygen content increases. Additionally, at high temperatures, the heating rate impact starts to disappear [133].

Moreover, Parthasarathy et al. [109] reported that the duration of residence in a given environment has an impact on char yield and its carbon content. They observed that increasing the residence period reduces char production while increasing char carbon content. By decreasing the char yield, a longer residence period allows for more time for the reactants in the volatiles to interact with the char and leads to a higher gas yield. Better devolatilization is achieved with a longer residence time, which increases the char's carbon content. Additionally, research has demonstrated that the BET surface area of the char has a direct correlation with the residence time. Initially, the BET surface area increases with the increase in residence time, but this effect levels off after prolonged times. This can be attributed to the sigmoidal-shaped curve of the devolatilization rate [134]. Zhang et al. [135] also observed that the BET surface area of chars increased with the residence time until a certain point, after which it began to decrease. This can be explained by the fact that chars' ability to generate pores might benefit from a fair extension of residence time at high temperatures. However, if the residence period is prolonged too much, the pore structure of the chars may be destroyed, which would then cause deactivation.

Lastly, the use of microwave absorbers to indirectly heat biomass particles during pyrolysis has been shown to increase the reaction temperature at relatively low microwave power. This increase in temperature has a significant effect on the yield and quality of the pyrolysis products [136]. In addition, the use of iron-based catalysts, such as ferric chloride or ferrocene, can help to produce high-quality graphene-like biochar with excellent physicochemical properties. Furthermore, by using ferrocene as a catalyst, it is possible to produce graphene oxide at a much lower temperature [62,68,137].

## 9. Formation Mechanism of Biochar during Pyrolysis

Recently, pyrolysis has been employed to produce GLC materials from a wide range of biomasses. In order to understand the mechanism of graphene formation during biomass pyrolysis, it is important to understand the mechanism of biochar formation, as the two mechanisms are closely related.

Most biomasses are composed of cellulose, hemicellulose, and lignin, which each degrade at different temperatures and through distinct pathways. For instance, the temperature range for the decomposition of hemicellulose is between 220 and 315 °C, cellulose is between 315 and 400 °C, and lignin is between 160 and 900 °C [138]. Examining the individual pyrolysis mechanisms of these components can provide further insight into the pyrolysis of biomass and the formation of GLC materials.

Figure 10 illustrates the formation mechanism of biochar from cellulose. The pyrolysis of cellulose is initiated by the depolymerization of cellulose into oligosaccharides and the subsequent breaking of glycosidic bonds to form D-glucopyranose. Intramolecular rearrangement of D-glucopyranose leads to the formation of levoglucosan, which can either be converted to levoglucosenone through dehydration or can be exposed to a combination of rearrangements and dehydrations resulting in the formation of hydroxymethylfurfural. Furthermore, levoglucosenone can be chemically altered in various ways, leading to the production of biochar. This includes dehydrating, decarboxylation, aromatizing, and undergoing intramolecular condensation. Additionally, hydroxymethylfurfural can break down to produce bio-oil and syngas that are more volatile, or alternatively become further polymerized, aromatized, and condensed to form biochar [49].

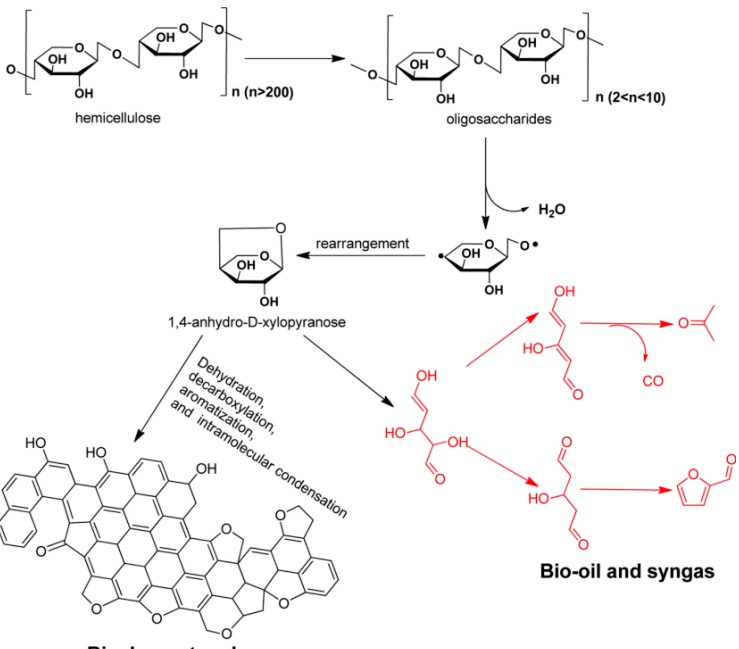

**Figure 10.** Formation mechanism of biochar by cellulose pyrolysis. Reprinted with permission from [49]. Copyright 2015 American Chemical Society.

The mechanism of hemicellulose pyrolysis is similar to that of cellulose pyrolysis. It is also depolymerized to form oligosaccharides, followed by glycosidic bond breakage and rearrangement of the depolymerized molecules to yield 1,4-anhydro-D-xylopyranose. Furthermore, it may be dehydrated, decarboxylated, aromatized, and intramolecularly condensed to create solid biochar, or it can break down into smaller compounds such as bio-oil, syngas, and low-molecular-weight compounds [49]. Figure 11 depicts the process of producing biochar from hemicellulose.

**Figure 11.** The formation mechanism of biochar by hemicellulose pyrolysis. Reprinted with permission from [49]. Copyright 2015 American Chemical Society.

The degradation of lignin is relatively intricate compared to cellulose and hemicellulose due to its more intricate structure, which is depicted in Figure 12. The principal mechanism in the pyrolysis of lignin is a reaction involving free radicals, which originate from the cleaving of the β-O-4 bond in the lignin molecules. The radicals are capable of scavenging protons from molecules with weak C-H or O-H bonds, leading to the production of breakdown products such as vanillin and 2-methoxy-4-methylphenol. As the reaction progresses, radicals are transferred to other species, resulting in a chain reaction. Ultimately, the chain reaction is halted when two radicals encounter each other and form a more stable compound. Nevertheless, since the detection of radicals in the pyrolysis process is exceptionally difficult, understanding the precise mechanism of lignin pyrolysis is a major challenge [49].

**Figure 12.** Lignin pyrolysis: biochar formation mechanism. Reprinted with permission from [49]. Copyright 2015 American Chemical Society.

## 10. Mechanism of GLC Materials Formation during the Biomass Pyrolysis Process

Most biomass is lignocellulosic and contains long chains of carbon, hydrogen, and oxygen compounds. The process of converting lignocellulosic biomass into graphene involves increasing the carbon content and arranging the carbon structures in a graphitic-like form. This process involves two steps: carbonization and graphitization. Carbonization involves the removal of light-molecular-weight compounds through heating, while graphitization is used to arrange the remaining carbon structures into a graphitic-like form. The converted carbon structure may not be similar to pure graphene, but the properties that they possess are somewhat graphene-like [44].

Debbarma et al. [64] synthesized graphene nanosheets from sugarcane bagasse via pyrolysis in the presence of sodium hydroxide. The chemistry behind the formation of graphene nanosheets is presented in Figure 13.

Sugarcane bagasse has a high concentration of cellulose, which is composed of glucose monomers held together by glycosidic bonds. During the pyrolysis of sugarcane bagasse, the breakdown of glucose monomers takes place, and these monomers contain aldehyde and hydroxyl groups. The hydroxyl group on the fifth carbon of the glucose molecule can then bind to the aldehyde group on the first carbon to form a cyclic hemiacetal structure. This structure is similar to pyran and consists of six-membered heterocyclic rings. It is thought that many of the glucose monomers were linked via glycosidic bonds during pyrolysis, and further condensation and aromatization of the cyclic rings occurred to form planar graphitic polyaromatic ring structures. Debbarma et al. [63] also synthesized GO from sugarcane bagasse where heating sugarcane bagasse at different temperatures caused the degradation of glucose monomers, leading to the formation of glycosidic bonds and polyaromatic rings. The presence of air facilitated oxidation, aromatization, and condensation, resulting in the formation of SBGO nanosheets. The mechanism is depicted in Figure 14.

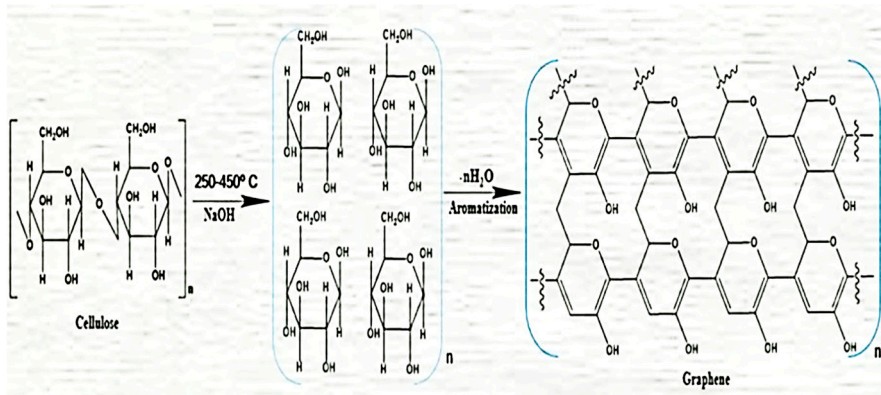

**Figure 13.** Mechanism of formation of graphene nanosheets at 250–350 °C. Reprinted from [64]. Copyright 2020, Debbarma, J.; Mandal, P.; Saha, M., used under Creative Commons Attribution License (CC BY) (https://creativecommons.org/licenses/by/4.0/ (accessed on 20 January 2023)).

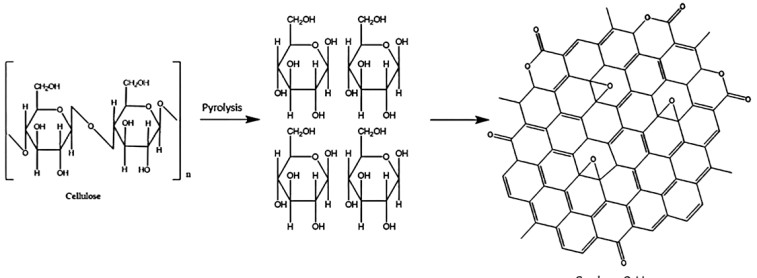

**Figure 14.** Mechanism of formation of GO from sugarcane bagasse [63]. Reprinted with permission from the publisher (Taylor & Francis Ltd., http://www.tandfonline.com (accessed on 20 January 2023)).

This same research group synthesized nitrogen-doped GO (N-GO), wherein the formation mechanism was the same as stated above, with primary amine groups from the amino acids enhancing the nitrogen content and resulting in the formation of N-GO, as illustrated in Figure 15 [139].

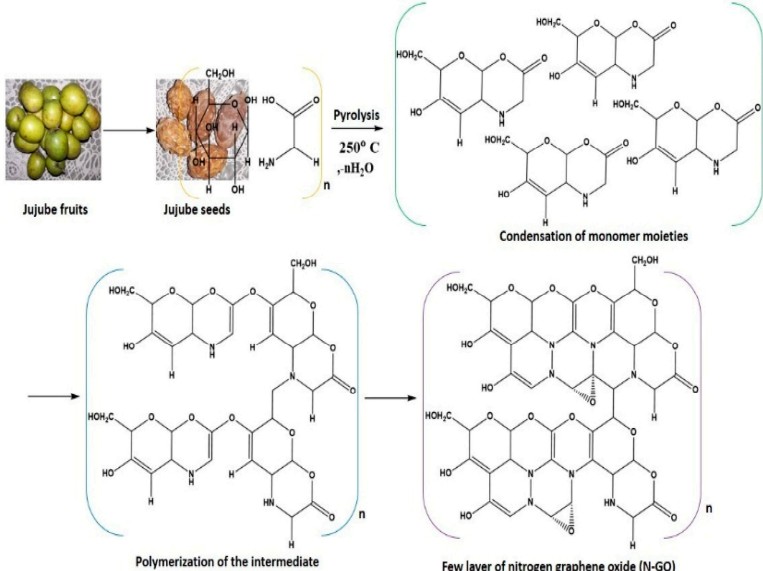

**Figure 15.** Mechanism of formation of N-GO [139]. Reprinted with permission from the publisher (Taylor & Francis Ltd., http://www.tandfonline.com (accessed on 20 January 2023)).

Roy et al. [76] developed a method for synthesizing graphene from tannic acid, alginic acid, and green tea through a controlled pyrolysis procedure, and proposed a formation mechanism of graphene from alginic acid, illustrated in Figure 16. They proposed that this reaction likely started with the production of radicals, which was then followed by the release of water molecules and $CO_2$ and the aromatization and intermolecular condensation reactions at a temperature of 1100 °C. They hypothesized that similar processes happened with the polyphenols from green tea and tannic acid based on the presence of carboxyl groups and vulnerable oxygen bonds in their molecular structures.

**Figure 16.** Pyrolysis reaction mechanism for graphene synthesis from alginic acid. Reproduced under the terms of CC BY-NC-ND license [76].

Omoriyekowan et al. [140] explored the process of carbon nanotube (CNT) formation in their most recent investigation, synthesizing CNTs with cellulose taken from PKS. In order to draw out the bio-components from the PKS, two distinct techniques were utilized. Once the extraction process had concluded, cellulose and lignin were then exposed to microwave pyrolysis. The end result of their research suggested that cellulose played a vital part in generating CNTs. Figure 17 illustrates the reaction pathways of the decomposition of cellulose to produce nanotubes. The authors examined bio-oils derived from lignin and cellulose to better understand the role of cellulose. Bio-oils derived from cellulose were high in monosaccharides, while bio-oils derived from lignin were rich in phenols and single-ring hydrocarbons. According to the authors, the breakdown of cellulose resulted in the production of monosaccharides such as D-glucopyranose, which was employed as a carbon source for CNT synthesis. Splitting of the glycosidic bonds in D-glucose generated anhydrides, oligosaccharides, and levoglucosan. Subsequently, these elements underwent degradation, cleavage, and rearrangement, leading to the formation of anhydro sugars and levoglucosan. The splitting of the C-O bonds in levoglucosan was followed by its aromatization, resulting in a formation of graphite layers, as shown in Figure 17.

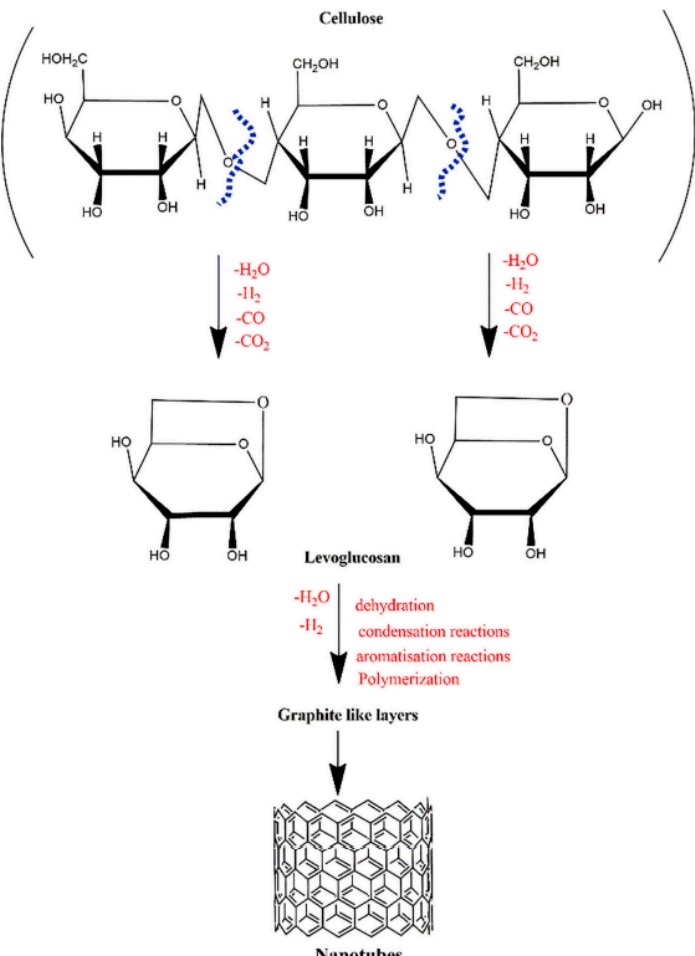

**Figure 17.** Reaction pathways during the decomposition of cellulose. Reprinted from [141], with permission from Elsevier.

Liu et al. [65] manufactured graphene using commercially accessible kraft lignin (KL) and carefully examined the formation process, structure, and features. They provided a general mechanism of lignin-based graphene that was catalyzed by iron, as demonstrated in Figure 18. The main reactions were pyrolysis and carbonization of KL at temperatures between 250 and 500 °C, wherein polyolefin compounds were converted into amorphous carbon (a-C) with the aid of iron particles via catalytic dehydrogenation. The a-C was in a metastable state and possessed a large amount of energy, so it required less energy to dissolve in iron metal than C atoms. The temperature required for carbon to dissolve into iron is 570 °C. Through the precipitation dissolution mechanism, a-C diffused into the metal particle and then precipitated as graphene on the free surface when the solid solubility limit was reached during cooling. Smaller metal particles and longer annealing times resulted in the migration of activated carbon species to the top surface and the nucleation of graphene. Graphene was observed when the holding time was in the range of 90–105 min, but had notably reduced areas and less graphene identified by Raman when the holding time was longer than 105 min. The probable cause of this was the growth of an $sp^2$ carbon network along the surface of the iron particles, which accumulated into a graphite shell. Additionally, the contact between the a-C and iron particles at higher temperatures resulted in a catalytic graphitization process. Iron has the ability to catalyze graphitization even at a low temperature due to the decomposition of iron carbide in the insulation stage.

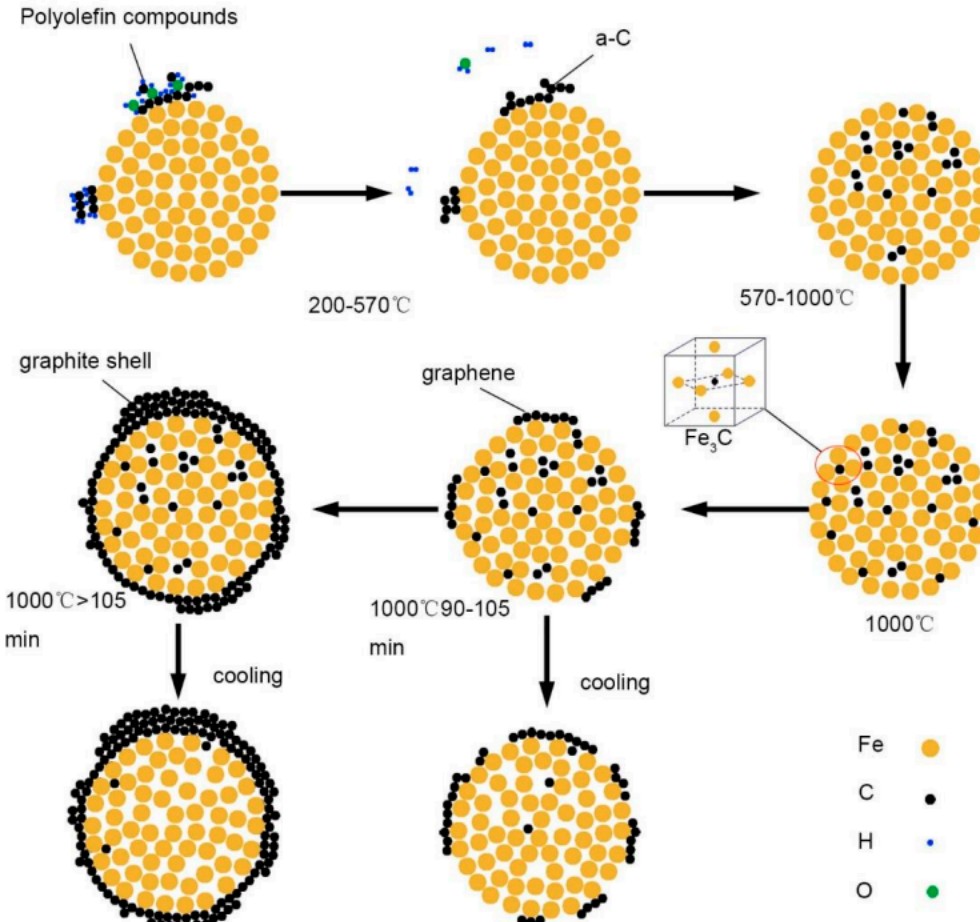

**Figure 18.** Schematic representation of the graphene synthesis processes from KL using iron particles as catalysts. Reprinted from [65] with permission from author.

The reaction equations were as follows:

$$Fe + C_a \rightarrow Fe_3C$$

$$Fe_3C \rightarrow Fe + C_g,$$

where $C_a$ is amorphous and $C_g$ is graphitic carbon.

The newly produced carbon from the breakdown of iron carbide is active and can be quickly transformed into graphite. On the other hand, too much iron may make the decomposition of $Fe_3C$ more challenging. The formation of graphene is carried out in two phases when iron particles are used as a catalyst. One of these is the precipitation and dissolution of carbon atoms, while the other is the manufacture and disintegration of iron carbide. As a consequence, the retention time has an influence on the formation of graphene.

Reviewing all the mechanisms discussed above, it may be concluded that the pyrolysis process involves the splitting and recombination of molecules. Carbon atoms form single covalent $sp^3$ bonds with other atoms, but during the graphene formation process, these bonds are broken, allowing the carbon atoms to form $sp^2$ bonds in the form of benzene rings. This process of nucleation leads to the development of graphene, as illustrated in Figure 19.

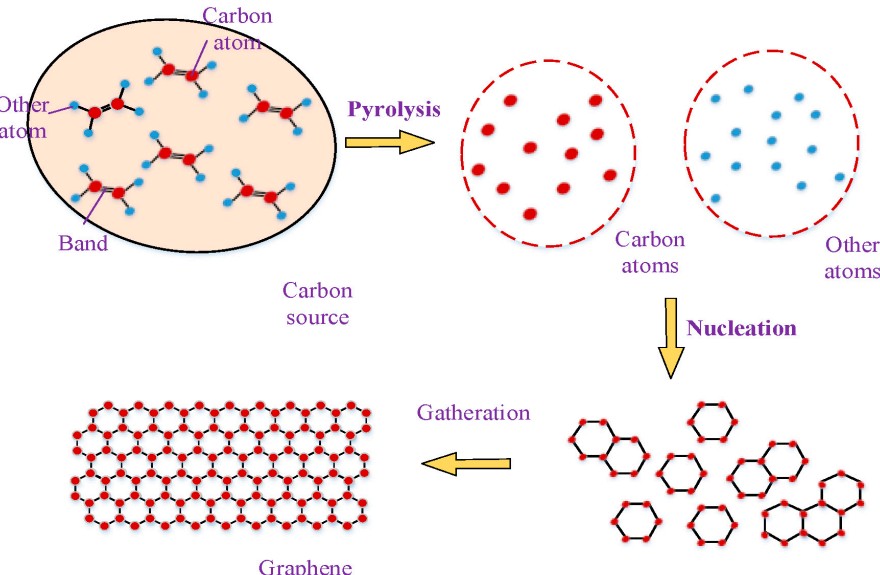

**Figure 19.** Mechanism of graphene formation via pyrolysis process [142]. Reproduced under the terms of CC BY 4.0 license.

## 11. Conclusion and Research Outlook

In conclusion, this review paper provides a comprehensive overview of the current state of knowledge of biomass-derived GLC materials and the microwave pyrolysis process for their synthesis. It was revealed that the microwave pyrolysis process is a promising solution for the cost-effective and renewable synthesis of GLC materials from biomass feedstock. Utilizing biomass waste to produce graphene can reduce high-expense production and associated pollution. Despite the fact that several studies have preferred high-temperature pyrolysis methods that use metal precursors along with biomass in order to break down the structure of biomass while also enabling volatile carbon materials to be deposited, future research may have to focus on utilizing lower-temperature thermal treatment to reduce the length of the reaction time. Even though bio-based graphene is not of the highest quality, the green synthesis route can still provide a good amount of multi-layer graphene, GO, and RGO. Further research is necessary to gain a more comprehensive understanding of the formation mechanism of GLC materials from biomass pyrolysis in order to optimize the production process, as well as to improve the efficiency of the microwave pyrolysis process. Additionally, the influence of feedstock particle size on the characteristics of the produced GLC material must be investigated. Furthermore, to gain a better understanding of the formation of GLC materials, individual pyrolysis of cellulose, hemicellulose, and lignin using the same process parameters should be conducted. With further research, biomass-derived GLC materials have the potential to become a viable and renewable alternative to traditional graphene materials for a variety of applications.

**Author Contributions:** Conceptualization, F.C.A. and G.C.S.; formal analysis, F.C.A.; writing—original draft preparation, F.C.A.; writing—review and editing, G.C.S.; supervision, G.C.S.; project administration, G.C.S.; funding acquisition, G.C.S. All authors have read and agreed to the published version of the manuscript.

**Funding:** The authors are thankful for the funding provided by the Natural Sciences and Engineering Research Council of Canada (RGPIN-2018-04440) and New Brunswick Innovation Foundation (NBIF-L2M2020-006).

**Institutional Review Board Statement:** Not applicable.

**Informed Consent Statement:** Not applicable.

**Data Availability Statement:** The raw/processed data required to reproduce these findings cannot be shared at this time as the data also form part of an ongoing study.

**Conflicts of Interest:** The authors declare no conflict of interest.

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
