# Peer review of "Graphene-like Carbon Structure Synthesis from Biomass Pyrolysis: A Critical Review on Feedstock–Process–Properties Relationship"

_carbon, 2023_

Round 1

Reviewer 1 Report

The authors tried to review the research progress of biomass-derived graphene-like carbon structure by pyrosis method.  Of course, this selected topic is very interesting, and will attract many readers' interest. However, there are some serious issues should be well addressed before its accpetance for publication.

1) This manusrcipt is not well organized.  For example:

2. Pyrolysis Process

2.1 Microwave Pyrolysis Reaction Mechanism

3. Current Trends on Synthesis of GLC Materials Via Biomass Pyrolysis Process

3.1 Suitable Biomass Feedstock for GLC Materials Synthesis Via Pyrolysis

5. Formation Mechanism

5.1 Mechanism of GLC Materials Formation During Biomass Pyrolysis Process

where are 2.2, 2.3, or 3.2. 3.2,3.3, 5.2, 5.3...?

2) Some paragraphs are written as Research article but Review paper. For example: Figure 4 illustrates the XRD spectrum of graphite, and the GO synthesized from orange peel, rice bran, sugarcane bagasse, and tri-composite agro waste respectively. From the XRD patterns it is evident that GO was successfully prepared only by TAW with the main diffraction peak located at 2θ = 12.705 and a particle size of 2.04 nm. The prominent and strong peak of GO (figure 4) indicates its excellent crystallinity, suggesting that GO pro- 245 duction was successful.

3)  The layout of the tables are poor. Table 1, 2, 3.

4) The Figure 6 is so simple, vague and superfacial.  The relationship between the parameters and the products is not well expressed.

5)   I don't think  6. Characterization Techniques is necessary.

6)  The English should be polished.  For example, 

this paper ultimately finishes by outlining the knowledge that still needs to be acquired and suggesting potential research paths to increase comprehension of the mechanism behind the formation of biomass-derived GLC materials.

In a word, this manuscript can't be accepted in the present form. 

Author Response

Comment #1:

This manuscript is not well organized.

Response: In response to the reviewer's comment, the manuscript has been revised and reorganized properly.

Comment #2:

Some paragraphs are written as Research article but Review paper. For example: Figure 4 illustrates the XRD spectrum of graphite, and the GO synthesized from orange peel, rice bran, sugarcane bagasse, and tri-composite agro waste respectively. From the XRD patterns it is evident that GO was successfully prepared only by TAW with the main diffraction peak located at 2θ = 12.705 and a particle size of 2.04 nm. The prominent and strong peak of GO (figure 4) indicates its excellent crystallinity, suggesting that GO pro-duction was successful.

Response:

Thank you for this insightful feedback. Authors have taken this into account and have now revised the manuscript with the correction of these types of paragraphs (highlighted in the manuscript).

Comment #3:

The layout of the tables are poor. Table 1, 2, 3.

Response:

The layout of the tables has been modified according to the manuscript template.

Comment #4:

The Figure 6 is so simple, vague and superfacial.  The relationship between the parameters and the products is not well expressed.

Response:

Figure 6 has been removed from the manuscript.

Comment #5:

I don't think  6. Characterization Techniques is necessary.

Response:

According to reviewers’ advice, “6. Characterization Techniques” section is removed.

Comment #6:

The English should be polished. For example,

“this paper ultimately finishes by outlining the knowledge that still needs to be acquired and suggesting potential research paths to increase comprehension of the mechanism behind the formation of biomass-derived GLC materials.”

Response:

In response to the reviewers' suggestion, the manuscript has been thoroughly revised and English language has been checked.

Reviewer 2 Report

According to my opinion, the manuscript entitled "Graphene-Like Carbon Structure Synthesis from Biomass Pyrolysis: A Critical Review on Feedstock-Process-Properties Relationship", given by Farhan Chowdhury Asif, Gobinda C. Saha can be published in Carbon, but after major  revision. Firstly, Introduction has to be extended, more up-to-date, should contain more elements from chemistry, more cutting edge graphene and nanographene application. Secondly, the manuscript lacks grahenes and nanographenes chemical structures. Such a review should be full of graphical presentations, for facilitating the comprehension of the work. Unfortunately, Fig 8, 9 and 10 presented only functionalised nanographene, but not graphene, which is presented in the form of nanotubes and nanosheets only on the scheme 11 and 12, respectively. Many figures presenting structures need to be added. I see also lack of comparison of various graphene and nanographene preparation methods. It is especially important taking into consideration that currently the bottom-up and APEX strategies are considered the most important and the most modern. One should be aware that biomass thermal degradation does not allow to obtain graphenes and nanographenes with controlled size, shape, functionalization and properties. Moreover, in what aspects can the method described in this review be competitive to bottom-up strategy? Finally, the most important issue, namely, no special “microwave effect” exists! Due to this fact, especially important is to present microwave-pyrolysis contrasted with pyrolysis in classical conditions and with the background of graphene and nanographene manufacturing.

Author Response

In light of above comments, following changes have been made in this revised manuscript (highlighted all changes):

- Introduction has been extended;

- More cutting edge applications are added;

- Chemical structures are added (Fig 2).;

- In the “Mechanism of GLC Materials Formation During Biomass Pyrolysis Process” section more figures are added presenting the structure (Fig 14,15, and 16);

- A brief comparison of various graphene synthesis methods has been added into the introduction section (paragraph 11-13). Also, it was mentioned that biomass thermal degradation does not allow to obtain graphene and nano graphene with controlled size, shape, functionalization and properties; and

- Two new sections (section 4 and 6) are added to discuss the special “microwave effect” as suggested by the reviewer.

Round 2

Reviewer 1 Report

This revised manuscript can be accepted in the present form. 

Reviewer 2 Report

I am fully satisfied with the corrected version of this manuscript.

Due to this fact, I propose to accept this review for publication in Carbon.